# Hereditary Tyrosinemia Type 1 Mice under Continuous Nitisinone Treatment Display Remnants of an Uncorrected Liver Disease Phenotype

**DOI:** 10.3390/genes14030693

**Published:** 2023-03-11

**Authors:** Jessie Neuckermans, Sien Lequeue, Paul Claes, Anja Heymans, Juliette H. Hughes, Haaike Colemonts-Vroninks, Lionel Marcélis, Georges Casimir, Philippe Goyens, Geert A. Martens, James A. Gallagher, Tamara Vanhaecke, George Bou-Gharios, Joery De Kock

**Affiliations:** 1Liver Therapy & Evolution Team, In Vitro Toxicology and Dermato-Cosmetology (IVTD) Research Group, Faculty of Medicine and Pharmacy, Vrije Universiteit Brussel, 1090 Brussels, Belgium; 2In Vitro Liver Disease Modelling Team, In Vitro Toxicology and Dermato-Cosmetology (IVTD) Research Group, Faculty of Medicine and Pharmacy, Vrije Universiteit Brussel, 1090 Brussels, Belgium; 3Faculty Health, Social Care and Medicine, Edge Hill University, St. Helens Road, Omskirk L39 4QP, UK; 4Department of Musculoskeletal & Ageing Science, Institute of Life Course & Medical Sciences, University of Liverpool, Liverpool L7 8TX, UK; 5Laboratoire de Pédiatrie, Hôpital Universitaire des Enfants Reine Fabiola (HUDERF), Université Libre de Bruxelles, Avenue Jean Joseph Crocq 1-3, 1020 Brussels, Belgium; 6Department of Laboratory Medicine, AZ Delta General Hospital, 8800 Roeselare, Belgium; 7Department of Biomolecular Medicine, Ghent University, 9000 Gent, Belgium

**Keywords:** liver disease, hereditary tyrosinemia type 1, nitisinone, transcriptomics, alkaptonuria, hepatocellular carcinoma, gene signature

## Abstract

Hereditary tyrosinemia type 1 (HT1) is a genetic disorder of the tyrosine degradation pathway (TIMD) with unmet therapeutic needs. HT1 patients are unable to fully break down the amino acid tyrosine due to a deficient fumarylacetoacetate hydrolase (FAH) enzyme and, therefore, accumulate toxic tyrosine intermediates. If left untreated, they experience hepatic failure with comorbidities involving the renal and neurological system and the development of hepatocellular carcinoma (HCC). Nitisinone (NTBC), a potent inhibitor of the 4-hydroxyphenylpyruvate dioxygenase (HPD) enzyme, rescues HT1 patients from severe illness and death. However, despite its demonstrated benefits, HT1 patients under continuous NTBC therapy are at risk to develop HCC and adverse reactions in the eye, blood and lymphatic system, the mechanism of which is poorly understood. Moreover, NTBC does not restore the enzymatic defects inflicted by the disease nor does it cure HT1. Here, the changes in molecular pathways associated to the development and progression of HT1-driven liver disease that remains uncorrected under NTBC therapy were investigated using whole transcriptome analyses on the livers of *Fah*- and *Hgd*-deficient mice under continuous NTBC therapy and after seven days of NTBC therapy discontinuation. Alkaptonuria (AKU) was used as a tyrosine-inherited metabolic disorder reference disease with non-hepatic manifestations. The differentially expressed genes were enriched in toxicological gene classes related to liver disease, liver damage, liver regeneration and liver cancer, in particular HCC. Most importantly, a set of 25 genes related to liver disease and HCC development was identified that was differentially regulated in HT1 vs. AKU mouse livers under NTBC therapy. Some of those were further modulated upon NTBC therapy discontinuation in HT1 but not in AKU livers. Altogether, our data indicate that NTBC therapy does not completely resolves HT1-driven liver disease and supports the sustained risk to develop HCC over time as different HCC markers, including *Moxd1*, *Saa*, *Mt*, *Dbp* and *Cxcl1*, were significantly increased under NTBC.

## 1. Introduction

Tyrosine inherited metabolic disorders (TIMD) are a subclass of inborn errors of amino acid catabolism, characterized by an inherited deficiency of a functional enzyme key for the metabolic pathway of tyrosine [1]. Tyrosine is broken down in fumarate and acetoacetate by a five-step enzymatic pathway that is mainly present in the liver and kidney cytosol. Each enzyme of this pathway is associated with one autosomal recessive inborn error [2,3].

Hereditary tyrosinemia type 1 (HT1, OMIM #276700) is the most severe and deadly TIMD with an overall incidence of 1 in 100,000 newborns worldwide [2,4,5]. In this case, the impaired enzyme of the tyrosine degradation pathway is the terminal enzyme fumarylacetoacetate hydrolase (FAH). Loss of FAH activity results in the accumulation of the upstream toxic intermediates fumarylacetoacetate (FAA), maleylacetoacetate (MAA) and succinylacetone (SA) (see Figure 1). These metabolites are responsible for the severe disruption of the intracellular metabolism of the liver and kidney. HT1 has a highly variable clinical presentation characterized by hepatic failure with comorbidities involving the renal and neurological system, which frequently results in death if left untreated [4,6,7]. In HT1, the liver is the most severely affected organ and a major cause of morbidity and mortality. The high risk of developing hepatocellular carcinoma (HCC) in HT1 patients is the most ominous and is the highest among all metabolic disorders [2,8]. Historically, a tyrosine- and phenylalanine-restricted diet has been applied as therapy for HT1 patients but has proven to be inadequate as it does not overcome the chronic complications, the production of the toxic intermediates or the development of HCC [9,10].

Since 1992, nitisinone or 2-(2-nitro-4-trifluoromethylbenzoyl)-1,3-cyclohexanedione (NTBC) has changed the clinical course and management of HT1 [9,11]. NTBC is a potent inhibitor of 4-hydroxyphenylpyruvate dioxygenase (HPD), resulting in an upstream block of FAH in the tyrosine degradation pathway preventing the accumulation of FAA, MAA and SA (see Figure 1) [12,13,14]. Although NTBC rescues HT1 patients from severe illness and early death, late complications can persist [15,16,17,18,19,20,21]. In order to manage the NTBC-induced impairments, a lifelong dietary adjustment by restriction of tyrosine and phenylalanine is applied for HT1 patients [10,22,23]. Despite the multiple benefits of NTBC, it does not restore the enzymatic defects inflicted by the disease nor does it cure HT1 [24]. Furthermore, the occurrence of HCC in some HT1 patients under NTBC therapy is described, meaning that there is still a certain residual activity in the liver even under NTBC treatment [4,25,26]. Although many efforts have already been made in understanding the origin of damages occurring in HT1, many cellular and molecular (oncogenic) mechanisms underlying the progression of HT1 still have to be unraveled [26]. 

In the present study, we want to identify changes in molecular pathways associated to the development and progression of the liver pathogenesis of HT1 that remain uncorrected under NTBC therapy by using gene expression profiling of *Fah*-deficient mouse livers under continuous NTBC therapy. As a control of a TIMD without hepatic manifestation, we used a homogentisate 1,2-dioxygenase (*Hgd*)-deficient mouse model of alkaptonuria (AKU, OMIM #203500) also under continuous NTBC treatment. AKU is a serious, autosomal recessive, multisystem disorder affecting 1 in 250,000 live births [12,13]. It results from a deficiency in the HGD enzyme responsible for the formation of MAA out of homogentisic acid (HGA). Loss of HGD function results in a blockage of the degradation of tyrosine and an accumulation of HGA in body fluids and urine [9,27,28,29]. The accumulated and circulating HGA will oxidize into benzoquinone acetic acid (BQA) and polymerize into a melanin-like pigment (see Figure 1), which preferentially deposits in connective tissues during a process called ochronosis. Ochronosis and thus AKU is characterized by premature arthritis, lithiasis, cardiac valve disease, fractures, muscle and tendon ruptures and osteopenia. Other systemic features include kidney, prostatic, salivary and gall bladder stones, renal damage or failure, respiratory complications and auditory impairment [12,14,24,30,31]. This makes AKU a chronically debilitating disorder with heterogeneous symptoms and although the tyrosine pathway is mainly localized in the liver, AKU is an extrahepatic disease with multifarious systemic symptoms. Since 2020, NTBC therapy is also the standard of care treatment for AKU patients.

Our study aimed to acquire fundamental knowledge on the uncorrected liver disease phenotype in continuously NTBC-treated HT1 patients. Our study focused on identifying the molecular mechanisms and pathways that remain modulated in the livers of *Fah*- and *Hgd*-deficient mice during continuous NTBC therapy and after NTBC discontinuation for seven days. To accomplish this, we performed microarray-based whole transcriptome profiling. We found that the most affected molecular pathways are those involved in liver-specific metabolic processes (synthesis/degradation), lipid homeostasis and hepatic cholestasis. Most importantly, we identified a set of 25 genes that discriminates HT1 livers from AKU livers, even under continuous NTBC therapy, as such representing remnants of an HT1-driven residual uncorrected liver disease phenotype.

## 2. Materials and Methods

### 2.1. Hereditary Tyrosinemia Type 1 Mouse Model 

The Fah^5981SB^ strain (referred to as HT1 mice), backcrossed on C57Bl/6J background and kindly provided by Markus Grompe (Oregon Health & Science University, Portland, OR, USA). The mice bear a single N-ethyl-N-nitrosourea-induced point mutation (G>A loss) leading to the splicing out of exon 7 within the Fah gene, resulting in a frameshift and subsequently the introduction of a premature stop codon at amino acid position 303. Consequently, the mice produce a truncated, unstable FAH protein that is degraded, making them a suitable model for HT1. If neonatal HT1 mice are not continuously administered NTBC, they die of acute liver failure [32,33]. Therefore, all mice, except during the withdrawal experiment, received through their drinking water continuous NTBC treatment (8 mg/L). To prevent the formation and accumulation of tyrosine and its toxic metabolites upon NTBC therapy discontinuation, the mice were provided ad libitum with an irradiated diet that was low in tyrosine and phenylalanine (LabDiet^®^ 5LJ5 chow, LabDiet, St. Louis, MO, USA), which resembles the protein-restricted diet of HT1 patients.

### 2.2. Alkaptonuria Mouse Model

The homozygous *Hgd* knockout-first allele mouse (Hgd tm1a^−/−^), also backcrossed on a C57Bl/6J genetic background, is used as an animal model of human AKU (referred to as AKU mice) [27]. AKU mice are kindly provided by George Bou-Gharios (Institute of Ageing and Chronic Disease, University of Liverpool, Liverpool, UK). Hgd tm1a^−/−^ mice contain an IRES:LacZ gene trap cassette and a promoter-driven neo cassette inserted into the fifth *Hgd* intron with the sixth exon flanked by loxP sequences. Homozygous Hgd tm1a^−/−^ mice, therefore, show an AKU phenotype based on *Hgd* gene disruption while heterozygous Hgd tm1a^−/+^ and homozygous Hgd tm1a^+/+^ mice show a normal wildtype phenotype. Hgd tm1a^−/−^ mice develop specific symptoms of AKU, including blackening of urine and progressive osteoarthritis when NTBC is not administered [27]. Therefore, similar to the aforementioned experimental conditions of the HT1 mouse model, all mice received through their drinking water continuous NTBC treatment (8 mg/L). An irradiated diet low in tyrosine and phenylalanine (LabDiet^®^ 5LJ5 chow, LabDiet, St. Louis, MO, USA) was provided. 

### 2.3. Animal Experiments

Animal procedures were conducted at the Vrije Universiteit Brussel and were approved by the Institutional Animal Ethics Committees (grant numbers 16-210-1 and 21-210-2). Mice were housed in group under SPF conditions in individual cages within a regulated environment of 19–23 °C and 30–70% R.H. with a 14/10-h light/dark cycle. To maintain their health, continuous NTBC treatment (8 mg/L) was given to both mouse models, and also to the pregnant females up until weaning, through their drinking water. The dosage used to treat both mouse models (HT1 and AKU) was based on the standard NTBC therapy dosage used to treat HT1 mice. As such, AKU mice can be used as a control to identify molecular changes in the liver of HT1 mice associated to the disease itself and not inflicted or masked by NTBC therapy. At eleven weeks of age, both mouse models (referred to as HT1-7dNTBC (*n* = 3) and AKU-7dNTBC (*n* = 4)) had their NTBC therapy withdrawn for seven consecutive days.

### 2.4. Sample Collection and Preparation

Sample collection was carried out according to the previously mentioned protocol [34]. Briefly, at 12 weeks of age, the mice of both experimental groups were anesthetized via intraperitoneal injection of a mixture of ketamine (87.5 mg/kg Ketamidor^®^ (Ecuphar, Catalonia, Spain)) and xylazine (12.5 mg/kg Rompun^®^ (Bayer, Leverkusen, Germany)). After a ventral heart puncture, blood was collected into ethylenediaminetetraacetic acid (EDTA)-coated microtubes (Sarstedt, Nümbrecht, Germany, K3E tube) and onto dried blood spot (DBS) cards (Whatman 903; GE Healthcare, Chicago, IL, USA). The blood samples were centrifuged at 1500× *g* for 15 min at 4 °C, and the serum was frozen at −80 °C until further use. To prepare liver tissue for transcriptome analysis, cubes with a maximum volume of 1 cm^3^ of liver tissue were collected in an RNA-protecting solution and stored at −80 °C. 

### 2.5. Dried Blood Spot Analysis

The DBS cards were analyzed as previously reported [34]. DBS cards were air-dried at room temperature for at least 24 h prior to analysis. An HPLC 1100 Agilent system coupled with AB Sciex API 3200 and API 4000 MS analyzers was used for analysis. In order to quantify Tyr (*m*/*z* 238 > 102) and Phe (*m*/*z* 222 > 102), small discs were perforated from the DBS cards and placed into a 96-microtiter plate. The DBS were eluted with methanol at room temperature for 20 min, and the resulting eluent was transferred into an additional 96-microtiter plate. Internal standards (IS) were added to separate wells. The dried residues from the original microtiterplate were used for SA quantification. A standard stock solution of labeled amino acids isotopes was added to every well of the additional microtiterplate. The latter was dried under nitrogen at 55 °C, dissolved in a mixture of Butanol-HCl, and incubated at 65 °C under an inert atmosphere. After the last evaporation step using a nitrogen flow, an acetonitrile-water-formic acid mixture was added as an eluting solution and the samples were analyzed by MS/MS via direct flow injection. Plasma Tyr and Phe concentrations were determined using labeled IS (^13^C_6_ Phe, *m*/*z* 228 > 102 and ^13^C_6_ Tyr, *m*/*z* 244 > 102).

SA (*m*/*z* 211 > 137) was quantified by exposing the DBS samples, the dried residues from the first plate, to a hydrazine hydrate solution containing an IS of deuterated SA (*m*/*z* 216 > 142). The mixture was incubated for 40 min at 50 °C after which the resulting samples were transferred to a new 96-microtiter plate and analyzed using MS/MS via direct flow injection. SA plasma concentrations were calculated using a standard curve. 

NTBC (*m*/*z* 330 > 218 and 330 > 126) concentrations were determined by eluting DBS discs with pure methanol and internal standard (^13^C_6_ nitisinone, *m/z* 336 > 218 and 336 > 126) for 30 min at room temperature, followed by direct use of the eluents in LC/MS. An isocratic LC method (0.5 mL/min) was used, with a Poroshell 120 EC-C_18_ column (Agilent, Santa Clara, CA, USA) and acetonitrile-water mixture (85:15) with 0.05% formic acid at 30 °C, to elute the sample solution. The plasma concentrations were calculated using a standard curve.

### 2.6. Extraction of Total RNA

RNA samples were collected in a mixture of RNAprotect Tissue Reagent (Qiagen, Hilden, Germany) and phosphate-buffered saline (5:1 *v*/*v*) to protect the RNA. The GenElute Mammalian Total RNA Purification Miniprep kit (Sigma-Aldrich, Bornem, Belgium) was used to extract total RNA from all samples in accordance with the manufacturer’s instructions. The extracted RNA was quantified using a Nanodrop spectrophotometer (Thermo Scientific, Wilmington, NC, USA).

### 2.7. Microarray Profiling and Analysis

For each experimental group, 100 ng RNA was extracted from liver tissue and amplified and in vitro transcribed using the Genechip Whole Transcriptome PLUS Reagent Kit according to the manufacturer’s instructions (Applied Biosystems, Waltham, MA, USA) as previously described [34]. The amplified RNA and synthetized single-stranded cDNA were purified using magnetic beads, followed by hydrolysis of 15 µg ss-cDNA using RNase H, fragmentation and labelling of, respectively, 5.5 µg and 3.5 µg ss-cDNA using Fragmentation Master Mix and Labelling Master Mix. The labeled cDNA was subsequently hybridized to the Affymetrix Mouse Gene 2.0 arrays and placed in a Genechip^®^ Hybridization Oven-645 (Affymetrix, Santa Clara, CA, USA) rotating at 14 g at 45 °C for 16 h. Post incubation, the arrays were washed on a Genechip^®^ Fluidics Station 450 (Affymetrix) and stained with an Affymetrix HWS kit as was indicated by the manufacturer. An Affymetrix GeneChip^®^ Scanner 3000 7G was used to scan the microarray chips. All chips were further subjected to quality control by using the Affymetrix GCOS software. The datasets were corrected, summarized and normalizes using Robust Multiarray Analysis. Transcriptome Analysis Console (TAC—version 4.0—Applied Biosystems) was used to create the heatmaps. Ingenuity Pathway Analysis software (version 2022-11) was applied to perform transcriptomic pathway analyses and gene set enrichment was determined based on a ≥2-fold difference and Benjamini–Hochberg (B-H) *p*-value ≤ 0.05. The data discussed in this manuscript have been deposited in the NCBI Gene Expression Omnibus and are accessible through GEO Series accession number GSE225001.

## 3. Results

### 3.1. NTBC Treated Fah-Deficient Mice Show Affected Molecular Pathways Involved in Liver-Specific Metabolic Processes, Lipid Homeostasis and Hepatic Cholestasis Compared to NTBC-Treated Hgd-Deficient Mice 

Under continuous NTBC treatment, HT1 and AKU mice showed comparable blood levels of NTBC (1.00 ± 0.23 µM and 0.91 ± 0.18 µM respectively) and tyrosine (545.4 ± 28.8 µM and 559.6 ± 71.7 µM respectively) (Figure 2A,B). However, a low level of residual SA was detected in the blood of HT1 mice (0.102 ± 0.035 µM) as opposed to AKU mice (not detectable) (Figure 2C).

Whole transcriptome profiling of HT1 and AKU liver tissue indicated that 64 genes were more than 2-fold upregulated and 29 genes more than 2-fold downregulated (*p*-value ≤ 0.05) in NTBC-treated *Fah*-deficient livers compared to NTBC-treated *Hgd*-deficient livers. Differentially regulated genes were enriched (2-fold, *p*-value ≤ 0.05) in toxicological gene classes related to liver disease (*Liver Hemorrhaging*), liver damage (*Degeneration of Liver*, *Liver Damage*, and *Liver Necrosis/Cell Death*), liver regeneration (*Liver Regeneration*, and *Liver Hyperplasia/Hyperproliferation*) and liver cancer (*HCC*). *Liver Hyperplasia/Hyperproliferation* (24 genes) and *HCC* (7 genes) are the most prominent toxicological gene classes with significantly modulated genes, as shown in Figure 3A,B.

Canonical pathway analyses showed that pathways involved in liver-specific metabolic processes (*FXR/RXR Activation*, *PXR/RXR Activation*, and *Aryl Hydrocarbon Receptor Signaling*), including the degradation and biosynthesis of amino acids (*tyrosine* and *asparagine*), hormones (*estrogen* and *melatonin*), neurotransmitters (*serotonin* and *catecholamines*), and other substances (*ethanol*, *acetone*, *nicotine*, *retinol* and *bupropion*), as well as lipid homeostasis (*Adpogenisis Pathway*, and *Acetate Conversion to Acetyl-CoA*), cancer (*SPINK1 General Cancer Pathway*, and *Circadian Rhythm Signaling*) and liver disease (*Hepatic Cholestasis*, and *Acute Phase Response Signaling*) were modulated (29 genes were downregulated and 64 genes were upregulated) in the livers of HT1 compared to AKU mice despite continuous NTBC therapy (Figure 4A,B). The modulated genes per pathway are represented in Appendix A.

Analysis of enriched upstream regulator sequences in these differentially regulated genes showed activation of signaling through interleukin (IL) 6 and tumor necrosis factor (TNF) (activation z-score ≥ 2; Figure 5A). Stearoyl-CoA desaturase (SCD) signaling, which regulates the expression of genes involved in lipogenesis and mitochondrial fatty acid oxidation, was inhibited (activation z-score ≤ −2; Figure 5A).

### 3.2. mRNA Marker Profile of the Uncorrected Liver Disease Phenotype in HT1 vs. AKU Liver and Differential Impact of NTBC Discontinuation

The genetic signature of the uncorrected remnants of the HT1-driven liver disease phenotype were defined. The gene expression of *Hgd* (65.3-fold), *Moxd1* (8.5-fold), *Mt1* (7.3-fold), *Mt2* (6.7-fold), *Saa2* (6.6-fold), *Dbp* (5.2-fold), *Cxcl1* (4.3-fold), *Asns* (3.7-fold), *Egr1* (3.6-fold), *Saa1* (3.1-fold), *Nr1d1* (3.0-fold), *Nr1d2* (2.5-fold), *Saa3* (2.2-fold), *Rbp1* (2.2-fold), *Lpl* (2.0-fold), *Nqo1* (2.0-fold) and *Abcg8* (2.0-fold) was significantly higher in *Fah*-deficient livers compared to *Hgd*-deficient livers under NTBC treatment (Figure 6A). In contrast, the gene expression level of *Fah* (−9.6-fold), *Elovl3* (−2.9-fold), *Arntl* (−2.6-fold), *Cyp3a41b* (−2.6-fold), *Acss3* (−2.3-fold), *Ddc* (−2.3-fold), *Fitm1* (−2.2-fold), *Cyp2c38* (−2.2-fold), *Slc22a7* (−2.1-fold) and *Nfil3* (−2.1-fold) was significantly lower in HT1 vs. AKU mouse livers under NTBC treatment (Figure 6A, Appendix A). Interestingly, when discontinuing NTBC therapy for seven consecutive days in AKU mice, only the gene expression of *Saa1* (−2.5-fold) was significantly modulated (Figure 6B). The upregulation of *Hgd* in *Fah*-deficient mice, and the downregulation of *Fah* confirms the gene disruption in both mouse models. As both mouse models are not complete knock-out models, but gene disruption models, some Affymetrix probes still bind truncated Fah and Hgd cDNA resulting in an ‘underestimation’ of the fold changes.

In contrast, when depriving HT1 mice from NTBC treatment for seven consecutive days, the gene expression of *Nqo1* (13.3-fold), *Cyp2c38* (3.5-fold), *Ddc* (3.1-fold) and *Mt2* (2.7-fold) significantly increased more than 2-fold, whereas the expression of the genes *Saa1* (−36.4-fold), *Elovl3* (−35.3-fold), *Saa2* (−22.7-fold), *Moxd1* (−6.3-fold), *Nr1d1* (−4.1-fold), *Cxcl1* (−2.8-fold) and *Dbp* (−2.7-fold) significantly decreased more than 2-fold (Figure 6C). 

Finally, when performing hierarchical clustering using these differentially expressed genes (excluding *Hgd* and *Fah*), a clear distinction was observed between HT1 and AKU mouse livers, independent of NTBC treatment. Furthermore, AKU livers treated or not with NTBC were clustered independently of the treatment condition, whereas for HT1 mouse livers, a clear difference was observed between continuously treated livers and those livers that underwent seven days of NTBC therapy discontinuation (Figure 6D).

## 4. Discussion

Since 1992, NTBC therapy has changed the clinical course and the well-being of HT1 patients. Despite its efficacy in preventing severe illness and early death, some HT1 patients may still experience late complications, such as the development of HCC, suggesting that there persists an uncorrected residual liver disease state even under continuous NTBC treatment. In order to characterize any potential uncorrected remnants of liver disease in HT1 patients under NTBC therapy, we performed whole transcriptome analyses of a preclinical mouse model of HT1 and compared it to a mouse model of AKU, another TIMD that does not harbor any hepatic manifestation. Both mouse models served the same genetic background and were kept under the same therapeutic conditions. The study revealed the upregulation of several genes involved in the development and progress of HCC, which could potentially serve as HCC markers.

Biochemical blood analyses of *Fah*-deficient HT1 mice under continuous NTBC therapy demonstrated residual levels of SA, compared to *Hgd*-deficient AKU mice in which SA could not be detected. This indicates that, although NTBC is a strong inhibitor of the HPD enzyme, it does not completely block the tyrosine degradation pathway, resulting in a residual enzymatic HPD activity in the liver. Although there are differences in metabolic rate, bioavailability, and the tyrosine catabolic pathway, 8 mg/L NTBC resembles the dose of 1–2 mg/kg/day for HT1 patients assuming the mice drink 3–5 mL water per day in conjunction with the Tyr- and Phe-restricted diet [23]. Nonetheless, higher NTBC concentrations can be used to obtain “complete” suppression of SA; however, mice can still develop HCC [35]. SA is associated with liver damage and with induced oxidative subcellular and tissue damage as SA leads to the accumulation of 5-aminolevulinic acid (ALA) [36]. When comparing the livers of HT1, harboring residual blood SA levels vs. AKU mice under continuous NTBC treatment, we observed that the differentially expressed genes group in toxicological gene classes associated with liver disease, liver damage, liver regeneration and liver cancer, in particular HCC. The most affected molecular pathways were those involved in liver-specific metabolic processes, lipid homeostasis and hepatic cholestasis. These observations point to a residual uncorrected liver disease state of HT1 mice under continuous NTBC therapy. Serum amyloid A (SAA), including SAA1, SAA2 and SAA3, are acute response proteins, mainly produced by hepatocytes and regulated by inflammation-associated cytokines, which promote endothelial dysfunction via a pro-inflammatory and pro-thrombotic effect and were detected to be significantly elevated in *Fah*-deficient livers under NTBC treatment. Their upregulation is a remnant of residual stress associated with residual SA levels. Its primary function is the regulation of the homeostasis. In chronic inflammation, the driving force in tumor development, SAA levels increase substantially as can be observed in this study. It has been reported that SAA’s augment the toxic effect of acetaminophen in liver tissue by promoting platelet aggregation on the cell membrane of liver sinusoidal endothelial cells [37].

Several genes involved in HCC development and prognosis were found to be significantly increased in liver tissue from NTBC-treated *Fah*-deficient HT1 mice compared to *Hgd*-deficient AKU mice. Indeed, *Asns*, also known as asparagine synthetase is an enzyme involved in the synthesis of asparagine. The expression of *Asns* has been observed to be elevated in HCC tumor tissues and closely correlates with serum α-fetoprotein (AFP) levels, tumor size, microscopic vascular invasion, as well as tumor encapsulation [38]. As such, *Asns* upregulation is a first indication that unresolved aspects of HCC development are still ongoing, even under NTBC therapy. This is further supported by the fact that the expression of metallothioneins (MT), *Mt1* and *Mt2*, are modulated in *Fah*-deficient mouse livers under NTBC treatment. MTs are small cysteine-rich metal-binding proteins that are crucial for metal homeostasis and protection against heavy metal toxicity, oxidative stress and DNA damage [39]. Recent studies have demonstrated that the abnormal expression of MTs, such as *Mt1* are able to trigger the process of carcinogenesis in various types of human malignancies, including HCC [40]. Collectively, MTs contribute to tumor metastasis by enhancing the invasion and migration of tumor cells and tumor microenvironment remodeling [40]. In the context of HCC, it was previously reported that the expression of *Mt1*, *Mt2* and metal transcription factor-1 (*Mft1*) is decreased in human HCC as compared with periportal-HCC and normal tissues [39]. Moreover, MTs have typical circadian rhythms and their expression depends on the differentiation status of the tumor [39]. Consequently, the increase in expression of *Mt1* and *Mt2*, found in Fah-deficient mouse livers under NTBC treatment, could point to an unresolved oxidative stress response against toxic tyrosine metabolites, which might still progress into HCC development. Interestingly, the expression of circadian clock target genes, including nuclear orphan receptor factor protein (*Nr1d1*) and *Nr1d2,* as well as D-box-binding protein (*Dbp*), was found to be upregulated in HT1 mouse livers under NTBC therapy. *Dbp* encodes a transcription factor that binds to the promoter of genes of albumin and several CYP enzymes. This is in accordance to what is found in HCC livers as compared with periportal-HCC and normal livers [39]. Furthermore, it has been reported that the early growth response protein *Egr1* and the copper-dependent monooxygenase *Moxd1*, which were found to be significantly increased in NTBC-treated HT1 livers, are correlated to the invasiveness of HCC cells and early tumor development, respectively [41,42]. Expression of *Moxd1* is also associated with poor survival in glioblastoma, whilst when it is downregulated, it activates ER-stress causing activation of the unfolded protein response. This latter has tumor-promoting functions [43].

NAD(P)H quinone oxidoreductase-1 (*Nqo1*) is a flavin-adenine dinucleotide (FAD)-dependent flavoprotein that catalyzes the reduction of quinones and their derivatives through the receptor NAD(P)H by loss of two electrons and as such avoid damage to cells [44]. We found that *Nqo1* expression is significantly increased in *Fah*-deficient mouse liver under continuous NTBC treatment. The overexpression of *Nqo1* has been observed in HCC and enhances the vulnerability of cells to oxidative stress-induced injury. *Nqo1* is also involved in regulating the proliferative and aggressive characteristics of HCC [45,46,47].

In the context of liver disease, we report here several modulated genes. More specifically, we found that *Abcg8* is significantly increased in HT1 livers under NTBC treatment. This is a cholesterol transporter in the liver and bile that operates with *Abcg5* as a heterodimeric transporter located at the canalicular membrane of hepatocytes and intestinal enterocytes where it actively transports sterols. Loss-of-function mutations in this gene are associated with an increased risk to develop gallstones. On the other hand, gain-of-function variants results in an increased function of the transporter and, as a consequence, increased biliary cholesterol levels [48]. Furthermore, *Rbp1*, a gene coding for retinol-binding protein 1, was also increased. *Rbp1* is involved in vitamin A metabolism and is highly expressed in hepatic stellate cells as well as in hepatic fibroblasts of fibrotic or cirrhotic livers [49]. In addition, *Cxcl1* encodes CXCL1 which is a major chemoattractant for neutrophils that binds to its receptor CXCR2 and was found increased in permanently treated HT1 mouse livers [50]. *Cxcl1* has an oncogenic role in HCC progression, as it is associated with tumor progression and recurrence in HCC patients [51]. Thereby it leads to the activation of signaling pathways such as PI3K/Akt, MAP kinases or phospholipase- β, resulting in the recruitment of neutrophils to inflamed areas. This neutrophil recruitment is also observed with the activation of the triggering receptor expressed on myeloid cells 1 (TREM-1), which promotes Akt activation [52,53]. Furthermore, it is also implicated in processes such as tissue repair and tumor development [52]. Notably, CXCL1 expression is elevated in the liver of non-alcoholic steatohepatitis (NASH) patients, but not in simple steatotic livers in obese individuals or in high-fat diet (HFD)-fed mice [53,54]. However, in a NASH mouse model induced by a choline-deficient amino acid-defined diet, increased *Cxcl1′s* hepatic mRNA levels in a toll-like receptor 4-MyD88-dependent manner are observed, causing an accumulated neutrophil infiltration associated with hepatic inflammation and fibrosis [55]. Interestingly, viral overexpression of CXCL1 in the liver is sufficient to trigger progression from steatosis to steatohepatitis in HFD-fed mice by inducing hepatic neutrophil infiltration, oxidative stress and hepatocyte apoptosis [56]. In contrast, several genes involved in lipid metabolism, including *Elovl3*, *Acss3* and *Fitm1*, were found to be significantly decreased in *Fah*-deficient mouse liver tissue under NTBC therapy. 

Importantly, when depriving *Fah*-deficient mice from NTBC for seven consecutive days, many of the aforementioned signature genes were modulated, which, however, was not the case in *Hgd*-deficient mice, supporting our conclusion that these genes are remnants from an unresolved HT1-drive liver disease state. The study thus revealed the upregulation of several genes that are involved in the acute phase and cancer development process, and which could serve as potential HCC markers, such as SAA, which is an early stage marker for acute (and chronic) inflammatory disease and CXCL1 is a prognostic indicator for poor outcome.

## 5. Conclusions

This study provides the first preclinical data on residual features of a possible unresolved HT1-driven liver disease state under NTBC therapy. Our study revealed numerous genes that are associated with liver disease and HCC that showed a differential expression in HT1 mouse livers vs. AKU mouse livers under continuous NTBC therapy. Specifically, we observed a significant increase in the expression of certain markers in the context of HCC development, some of which were further modulated upon NTBC therapy discontinuation. Altogether, we propose here a unique liver disease signature for HT1 under NTBC treatment comprising 25 genes (excluding *Fah* and *Hgd*), which indicates that NTBC therapy does not necessarily completely resolves HT1-driven liver disease or completely abolishes the risk to develop HCC over time.

## Figures and Tables

**Figure 1 genes-14-00693-f001:**
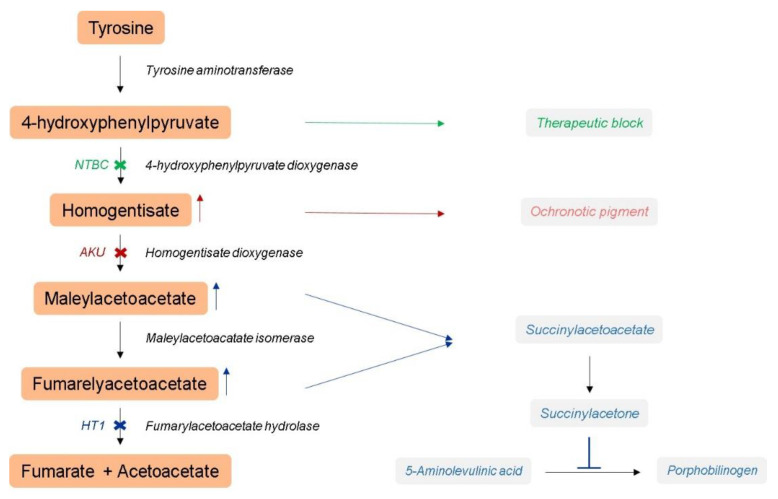
Schematic representation of tyrosine degradation in the liver. HT1 patients lack a functional FAH enzyme that causes the build-up of toxic tyrosine degradation products MAA and FAA, as well as the production of SA through an alternative degradation route. In AKU patients, the HGD enzyme is defective causing the accumulation of the toxic tyrosine metabolite HGA. NTBC provides an effective metabolic block by inhibiting the upstream HPD enzyme. This circumvents the accumulation of the aforementioned toxic tyrosine intermediates, rescuing patients from severe illness and even death. Abbreviations: HT1—Hereditary tyrosinemia type 1; FAH—fumarylacetoacetate hydrolase; MAA—maleylacetoacetate; FAA—fumarylacetoacetate; SA—succinylacetone; AKU—alkaptonuria; HGD—homogentisate dioxygenase; HGA—homogentisic acid; NTBC—2-(2-nitro-4-trifluoromethylbenzoyl)-1,3-cyclohexanedione; HPD—4-hydroxyphenylpyruvate dioxygenase.

**Figure 2 genes-14-00693-f002:**
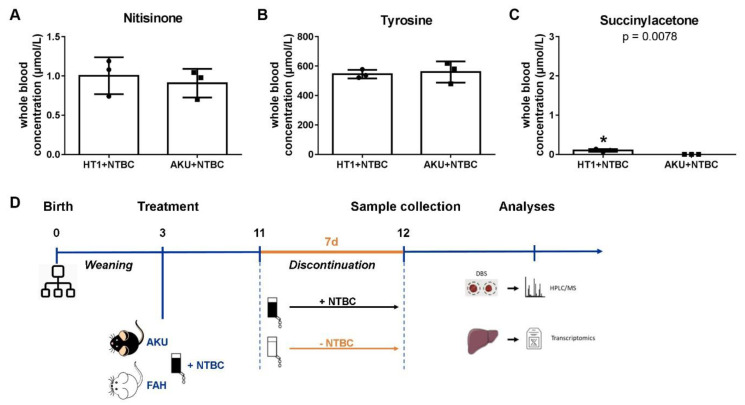
Biochemical evaluation of the HT1 and AKU mouse models. Quantification of (**A**) NTBC, (**B**) tyrosine and (**C**) succinylacetone blood levels. Data are represented as mean ± SD with * *p* < 0.05. (**D**) Schematic representation of the comparison experiment and subsequent blood and liver sample collection.

**Figure 3 genes-14-00693-f003:**
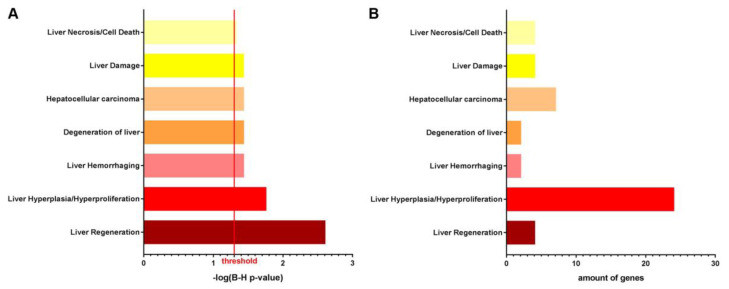
Whole transcriptome analyses of HT1 vs. AKU mouse liver tissue under continuous NTBC therapy. Toxicological gene class grouping of 2-fold up- and downregulated genes showing (**A**) B–H *p*-value of overlap and (**B**) amount of modulated genes.

**Figure 4 genes-14-00693-f004:**
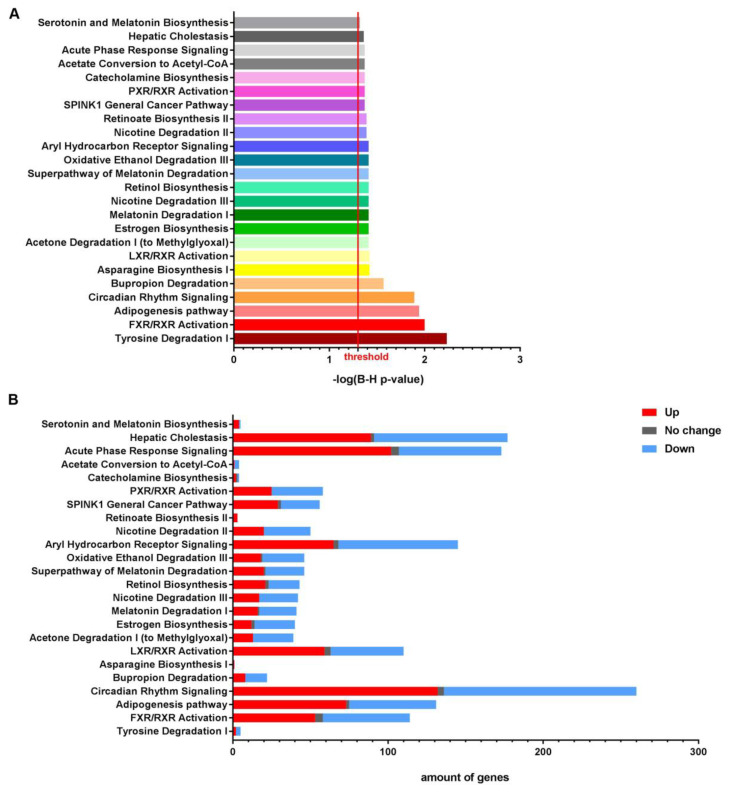
Canonical pathway analyses of whole transcriptome data comparing HT1 vs. AKU mouse livers under continuous NTBC treatment, showing (**A**) B–H *p*-value of overlap and (**B**) amount of modulated genes.

**Figure 5 genes-14-00693-f005:**
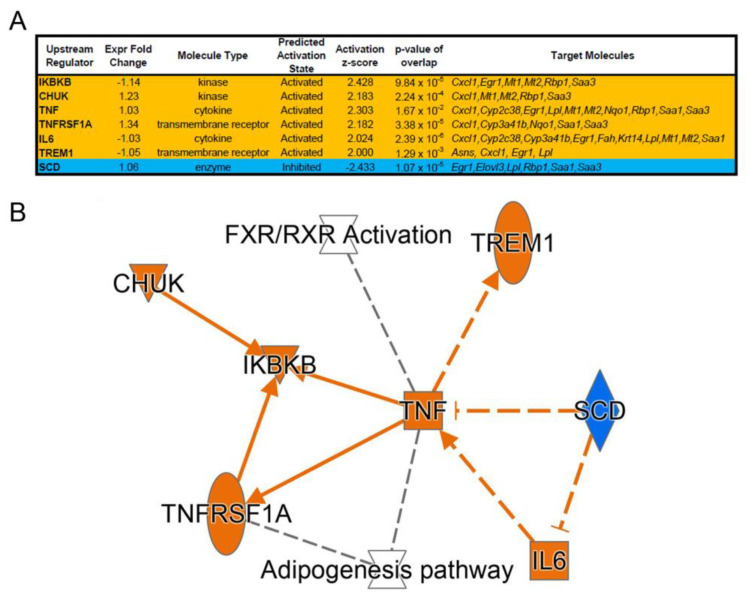
Upstream regulator analyses of whole transcriptome data comparing HT1 vs. AKU mouse livers under continuous NTBC therapy, showing (**A**) activation states and target molecules and (**B**) interactive network.

**Figure 6 genes-14-00693-f006:**
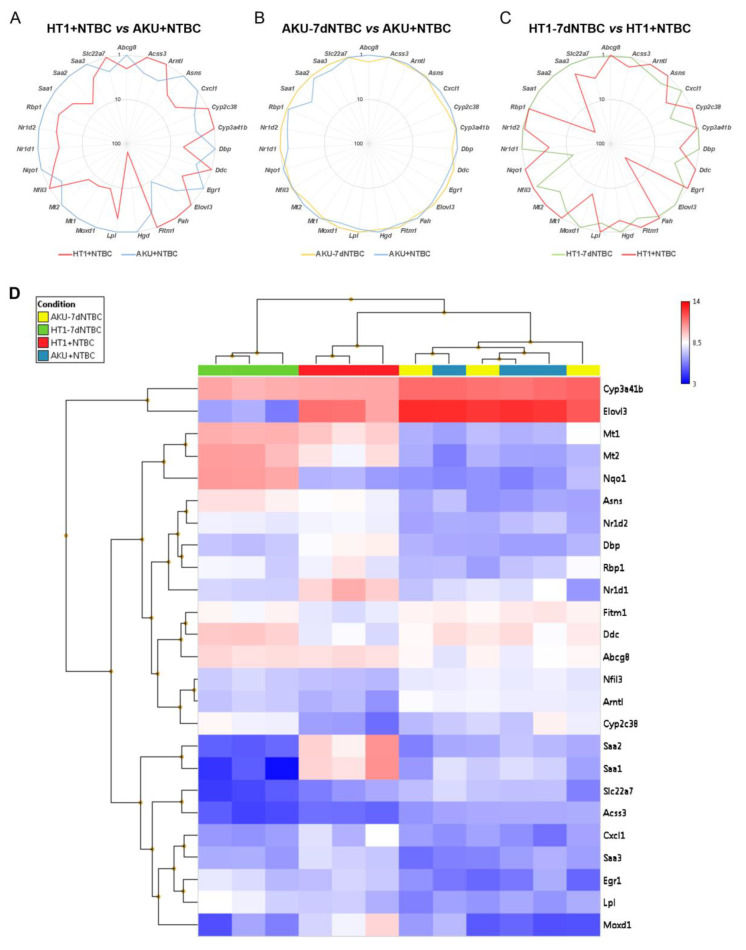
Genetic signature representing the uncorrected HT1-driven liver disease phenotype. Spider graphs showing differentially-expressed genes between (**A**) HT1 and AKU mouse livers under continuous NTBC therapy (HT1 + NTBC vs. AKU + NTBC), (**B**) AKU mouse livers upon seven days of NTBC discontinuation (AKU-7dNTBC) vs. continuously treated (AKU + NTBC) and (**C**) HT1 mouse livers upon seven days of NTBC discontinuation (HT1-7dNTBC) vs. continuously treated (HT1 + NTBC). (**D**) Hierarchical clustering of HT1 and AKU mouse livers with and without NTBC therapy using the generated genetic signature of the uncorrected HT1-driven liver disease phenotype (excluding *Hgd* and *Fah*).

## Data Availability

The data presented in this study are openly available in in the NCBI Gene Expression Omnibus and are accessible through GEO Series accession number GSE225001.

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
