# Peer review of "Hereditary Tyrosinemia Type 1 Mice under Continuous Nitisinone Treatment Display Remnants of an Uncorrected Liver Disease Phenotype"

_genes, 2023, doi:10.3390/genes14030693_

Round 1
Reviewer 1 Report
Dear Editor,
In this study, Neukormans et al. conducted a very successful the effects of nitisinone treatment on liver disease in mice with tyrosinemia type 1.
Overall, this study is well organised and well written
Here are my comments on the article.
1- In this study, the authors indicate that nitisinone treatment does not inhibit liver pathology in tyrosinemia type 1. Although nitisinone treatment has been widely used in patients with tyrosinemia type 1 for years, the success of nitisinone treatment is directly related to the timing of treatment initiation. As we know, the incidence of HCC is lower when NTBC treatment is started before the first year of life. When NTBC treatment starts before 1 year of age, the incidence of HCC is lower than 1%; when NTBC treatment starts between 1and 2 years of age, the incidence of HCC is 7%; and when NTBC treatment starts between 2and 7 years of age, the incidence of HCC is 21%. This study does not provide information on the subtypes of tyrosinemia.
2 - Two study groups (with tyrosinemia type 1 and alkaptonuria) both receive the same NTBC doses, but in reality tyrosinemia patients and alkaptonuria patients receive different NTBC doses, some tyrosinemia patients receive 10 or 20 times that of alkaptonuria patients. It is not reasonable to compare these treatment regimens. In addition, the NTBC dose may not be sufficient to inhibit the levels of SA in mice with tyrosinemia type 1.
Author Response
Please see the attachment.
Kind regards,
Jessie Neuckermans

Reviewer 2 Report
This manuscript describes an original and comprehensive study. It provides the first preclinical data on residual features of a possible unre-443 solved HT1-driven liver disease state under NTBC therapy and identified the genes associated with the liver disease and HCC in HT1. I think this study is going to have an important contribution to the literature.
Reviewer 3 Report
In their manuscript, Neuckermans and colleagues examined the expression of genes in NTBC-treated mice with hereditary tyrosinemia type 1 to examine factors that may contribute to development of hepatocellular carcinomas despite NTBC treatment, as compared to a control of another tyrosine metabolism disorder, alkaptonuria, which does not present liver abnormalities clinically. The identified a subset of 25 genes related to liver pathology which were differentially expressed, including some with a differential response to NTBC in the HT1 mice but not in the AKU mice.
This certainly addresses an important issue in HT1, regarding long-term outcomes and surveillance of NTBC-treated HT1-affected individuals. The manuscript clearly presents interesting and novel findings. The only comments I do have are minor in nature:
1. Were the pregnant females treated with NTBC? If that was the case, was it administered prenatally for both HT1 and AKU mice?
2. I would suggest that the authors better address the inherent limitations of their study. Fox example, in the second paragraph of their discussion, the authors discuss that given the presence of residual succinylacetone, this demonstrates that NTBC does not completely block the enzyme. There are however some limitations to this, including that the dose of NTBC given to mice is subject to the quantity of water that they drink, and therefore can be inconsistent. Complete suppression of succinylacetone has been seen in mice with higher doses of NTBC and low tyrosine diet (Al-Dhalimy et al. MGM 2002); it is to be noted that despite this, one mouse in this group still developed hepatocarcinoma. The limitations probably do not affect significantly the author’s conclusions, but the authors should still consider acknowledging them.
Overall, the work presented by the authors is of great quality, and I certainly hope to read some more of their work in the future.
Author Response

(The authors gave the same response as above.)

Round 2
Reviewer 1 Report
Dear editor,
The authors' responses to my previous reviews were acceptable, so this manuscript can be accepted for publication